# Extinction Effect of Foliar Dust Retention on Urban Vegetation as Estimated by Atmospheric PM10 Concentration in Shenzhen, China

Tianfang Yu [1], Junjian Wang [2,3], Yiwen Chao [1] and Hui Zeng [1,*]

1 Peking University Shenzhen Graduate School, Peking University, Shenzhen 518055, China
2 State Environmental Protection Key Laboratory of Integrated Surface Water-Groundwater Pollution Control, School of Environmental Science and Engineering, Southern University of Science and Technology, Shenzhen 518055, China
3 Guangdong Provincial Key Laboratory of Soil and Groundwater Pollution Control, School of Environmental Science and Engineering, Southern University of Science and Technology, Shenzhen 518055, China
* Correspondence: zengh@pkusz.edu.cn; Tel.: +86-0755-26035585

**Abstract:** Foliar dust retention is a crucial source of uncertainty when monitoring the vegetation index using satellite remote sensing. As ground sampling conditions are limited by vegetation dust retention, separating the extinction effect of foliar dust retention from the normalized difference vegetation index (NDVI) poses a significant challenge. In this study, we conducted a correlation test between the relative change in NDVI (δNDVI, an indicator of extinction effect) retrieved by the Gaofen-4 satellite and the atmospheric PM10 concentration in different meteorological periods (before, during, and after rainfall) across 14 stations in Shenzhen City, China. The results showed a significant correlation between δNDVI and atmospheric PM10 concentration during the before-rainfall period and weaker correlations for the other periods ($R = 0.680$, $p < 0.001$, $n = 63$ when excluding the during- and after-rainfall data). The correlation was more significant for the stations with low NDVI values, and a coastal station had a distinct regression slope of δNDVI versus PM10 from the other stations, indicating that the extinction effect of foliar dust retention in high-NDVI and coastal areas may not be well predicted by the general δNDVI–PM10 relationship. This provides a new quantitative basis for estimating the extinction effect of foliar dust retention using PM10 data for future improvement of the accuracy of vegetation monitoring by remote sensing.

**Keywords:** foliar dust retention; vegetation index; Gaofen-4 satellite; Shenzhen City; remote sensing application

## 1. Introduction

Foliar dust retention occurs when leaves trap particles from the atmosphere, forming a dust layer on the leaf surface. Numerous studies have shown that mature plants capture dust mainly through tissues on the leaf surface [1,2], such as grooves [3], trichomes [4], stomata [5], and leaf waxes [6]. Foliar dust can enter surface runoff under specific rainfall erosion intensity conditions or become resuspended when the wind blows [7]. For example, Yang et al. [8] demonstrated that in 2002, vegetation dust retention in central Beijing removed 772 tons of fine particulates from the atmosphere; while Tallis et al. [9] estimated that urban canopies in Greater London could remove 852-2121 tons of fine particulates per year. These findings suggest that dust-retaining urban plants play an important role in trapping atmospheric particulate matter.

The specific spectral information is controlled by multiple factors, including the original spectral characteristics of the leaves, dust retention capacity, and dust retention components [10,11]. However, numerous observations performed with ground object spectrometers have shown that when leaves are affected by foliar dust, their reflectivity

generally increases in the visible light band and decreases in the near-infrared band [12]. The change in spectrum absorption caused by dust on leaves is known as the extinction effect of foliar dust retention (EEFDR) [13].

The optical interference of EEFDR is significant. After analyzing the relationship between pigment concentration and reflectance spectrum on different levels of foliar dust retention at a leaf scale, Lin et al. [14] reported that dust deposition could affect the correlation between vegetation chlorophyll and spectrum. Using a double beam spectrophotometer to measure reflectance spectra on leaves of Ficus microcarpa, Xu and Yu [15] found that the reflectance of dirty leaves was overestimated by approximately 6.6% in the visible light spectrum and underestimated by approximately 25.6% in the near-infrared spectrum. The counterpart reported by Peng [16] on pear leaves is 7.75% and 10.23%, respectively. These results indicate that if EEFDR is ignored, the NDVI and other vegetation indices (VIs) obtained by satellite remote sensing data inversion will be significantly underestimated. VIs are important optical parameters reflecting vegetation status and are widely used in remote sensing observations [17]. Therefore, it is essential to correct the optical effects induced by foliar dust retention, as this will significantly improve the accuracy of VIs determined via remote sensing inversion.

Although foliar dust retention changes the remote sensing spectrum, it is currently impossible to invert the EEFDR in remote sensing using foliar dust retention because of the expense of artificial sampling and the difficulty in promptly updating remote sensing observations. In addition, the obstacles to retrieving EEFDR by satellite remote sensing also include cross-scale simulation, ground experiment complexity, dynamic observation conditions, and complicated dust retention conditions [18–21]. As a result, the state of foliar dust retention cannot be obtained widely and continuously. The extensive gap in determining how EEFDR affects satellite remote sensing observations needs to be filled.

Atmospheric PM10 concentration data has an obvious theoretically positive correlation with foliar dust retention [22]; thus, PM10 data is a potential substitute for foliar dust retention. PM10 data can be acquired continuously, stably, and at a low cost, which makes it suitable for remote sensing inversion. Therefore, we believe that atmospheric PM10 concentration can be used as a proxy index of foliar dust retention.

Considering the variations of NDVI and the variability of foliar dust retention with meteorological changes, the revisit time of the satellite should be as short as possible. Furthermore, Yan [13] argues that the low spatial resolution of MODIS cannot exactly match the leaf sampling sites because mixing pixels will cause errors. Therefore, we expected that using the Gaofen-4 satellite with high spatial and temporal resolutions instead of MODIS would allow for relatively accurate monitoring of the variations in EEFDR with PM10.

This study detected and analyzed the systematic bias caused by EEFDR to the vegetation indices retrieved from satellite remote sensing images, using the daily atmospheric PM10 concentration dataset and daily meteorological dataset in Shenzhen City, China. The study aims to: (1) quantify information on the potential correlation between δNDVI—an indicator of EEFDR, and atmospheric PM10 concentration—an indicator of foliar dust retention; (2) identify the space-dependency of the correlation based on site-specific NDVI; and (3) identify the time-dependency of the correlation based on different meteorological scenarios.

## 2. Materials and Methods

### 2.1. Study Region

Shenzhen City, a mega city in South China (113.7°–114.6°E, 22.4°–22.8°N), was selected as the study region. Shenzhen has a subtropical monsoon climate. The city's total area is approximately 2000 km². The vegetation types in the urban area are diverse and complex, but primarily dominated by evergreen broad-leaved forests. Figure 1 shows the study region's location and land cover types. Since China's opening-up policy was implemented, Shenzhen has been rapidly urbanized and industrialized, resulting in rapid changes in the land use structure and pattern. Specifically, large amounts of agricultural land have been,

and continue to be, transformed into various types of urban development areas, which has drastically increased the tendency toward vegetation distribution fragmentation. Most of the existing natural vegetation depicts an island-like distribution, and only the eastern coastal areas have roughly continuous vegetation banded distribution.

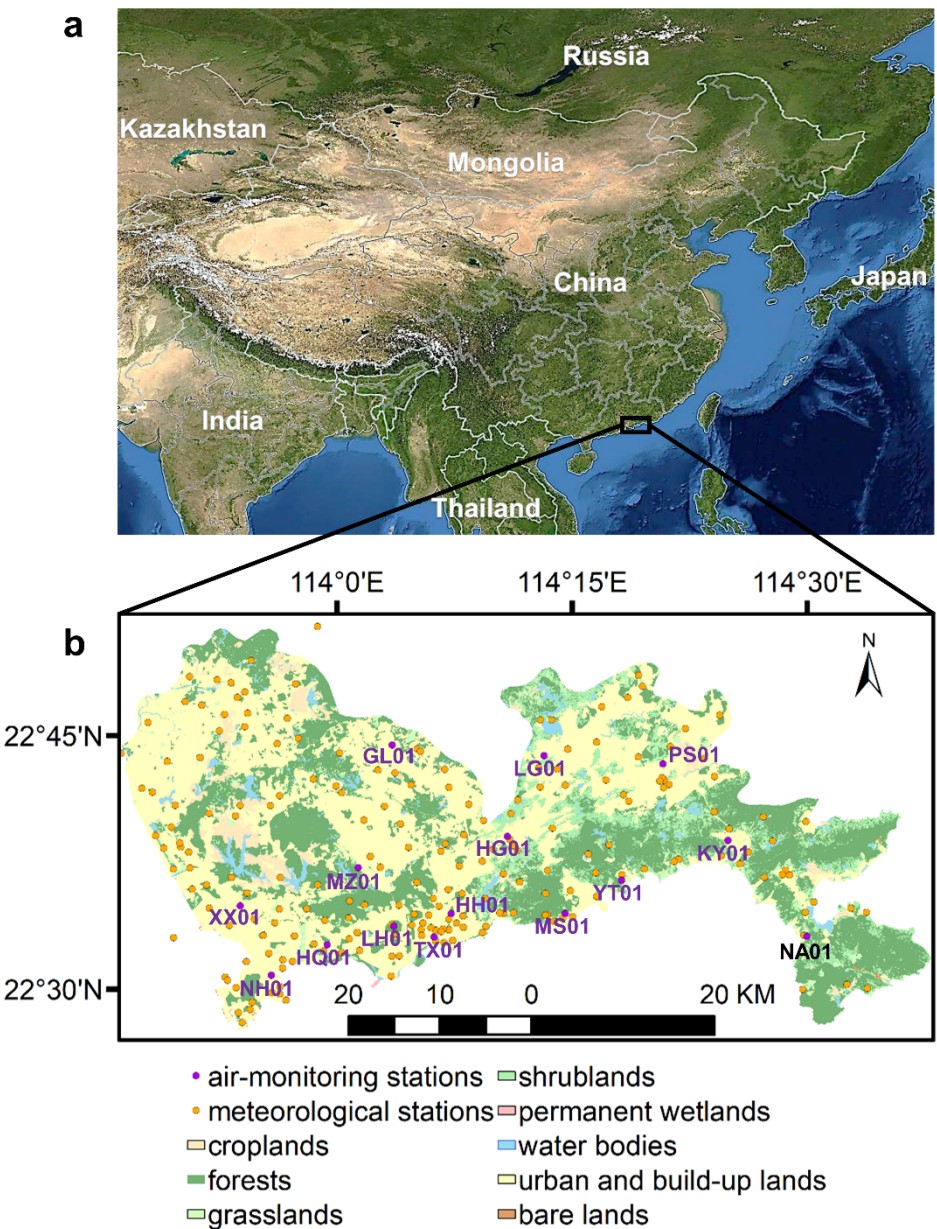

**Figure 1.** Geographical location of the study area. (**a**) Diagram of the geographical location of Shenzhen; (**b**) Land cover of the study region and distribution of air pollution monitoring stations (purple, *n* = 15) and meteorological stations (orange, *n* = 209). The air pollution monitoring station, Nan'ao station (NA01, black text), is unavailable and excluded.

*2.2. Ground Datasets*

To separate the dust retention signal from the phenological influence signal, the growing season must be chosen as the research period (i.e., from January–September), during which the vegetation index would not decrease due to phenology. Meanwhile, the rainy season with frequent cloudy and rainy weather occurs from April to September in Shenzhen and is not conducive to foliar dust accumulation or observation via optical sensors.

Considering the availability of data and the monthly variation trend of NDVI, we selected 1 January 2021 to 1 April 2021 as the research period, because it is characterized

by low rainfall frequency, as well as a clear and cloudless growing season. Ten images obtained by the Gaofen-4 satellite served as the primary data source for this study. Previous studies have reported that the NDVI in South China, where Shenzhen is located, has been rising slightly over the same period of years [23], indicating that the declining NDVI trend is not caused by phenology.

Figure 2 shows the dates of observations and rainfall events during the whole research period, along with the estimated precipitation amounts (interpolated by Kriging) of three rainfall events for each station. The ten observation dates were 1 January, 12 January, 16 January, 29 January, 4 February, 7 February, 15 February, 22 February, 6 March, 15 March, and 26 March, respectively. Herein, we used periods 1–9 to represent the 9 time intervals between all 10 adjacent remote sensing observations. According to the meteorological information, we divided them into periods 1–5 (before rainfall), periods 6–8 (during rainfall), and periods 7–9 (after rainfall), as shown in Figure 2.

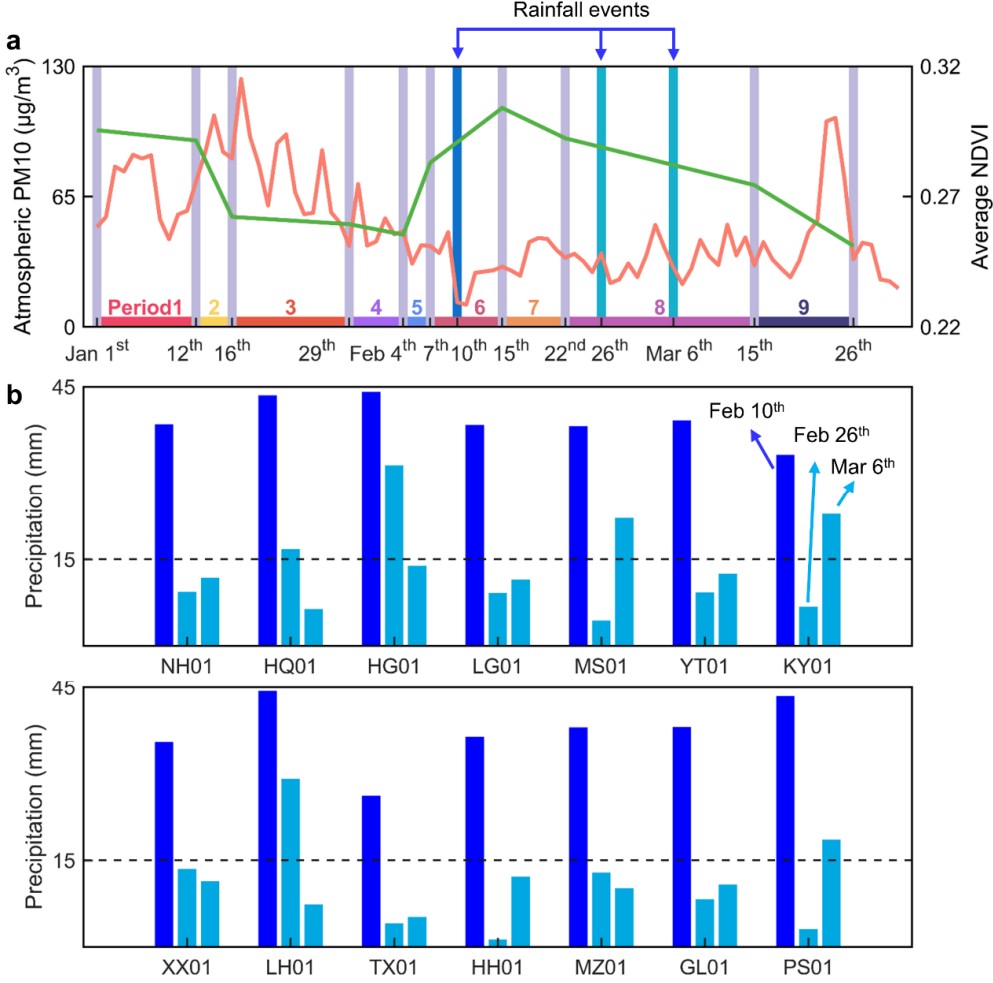

**Figure 2.** Overview of observational events and rainfall events. (**a**) Orange and green lines show the daily variations in average PM10 concentration and the normalized difference vegetation index (NDVI) of all stations, respectively. Lavender bars = 10 days of remote sensing data collection. Since the availability of remote sensing satellite imagery is limited by cloud cover, all sampling days exhibited clear skies; dark blue bars = rain on 10 February; light blue bars = rains on 26 February and 6 March. The colored bars during the two collection days represent periods 1–9. (**b**) Simulated precipitation (interpolated by Kriging) for the three rainfall events at each station in the study period. Dark blue bars = rain on 10 February; light blue bars = rain events on 26 February and 6 March. A boundary line of 14mm of precipitation is marked (which is considered the lowest limit for complete removal of foliar dust).

The ground datasets consisted of two parts: the dataset of atmospheric PM10 concentration and the meteorology. The dataset of the atmospheric PM10 concentration was provided by 14 government-controlled air-monitoring sites in Shenzhen, and the data were released daily. According to the Technical Rules for Selection of Ambient Air Quality Monitoring Stations, the height of monitoring ports of all automatic monitoring stations is between 3 and 20 meters from the ground. Table 1 shows the abbreviations, latitude and longitude, dominant vegetation type, and precipitation of each station. The foliar dust retention particle size mainly ranged from 10–50 μm [24]. Compared with PM2.5, PM10 showed a more significant correlation with vegetation less susceptible to meteorological conditions [25]; therefore, it was selected as the main indicator. The dataset of meteorology included data on rainfall and atmospheric visibility, which were released daily from 209 meteorological stations from the Meteorological Bureau of Shenzhen Municipality [26] in Shenzhen. The spatial distribution of both the PM10 and meteorological data is shown in Figure 1. In this study, the Kriging interpolation method was used to interpolate daily rainfall information into the whole Shenzhen region.

**Table 1.** Rainfall over 3 periods for 15 stations, interpolated by Kriging.

| Name | Abbreviation | Longitude | Latitude | Dominant Vegetation Type | Precipitation (mm) | | |
|---|---|---|---|---|---|---|---|
| | | | | | **10 February** | **26 February** | **6 March** |
| Nanhai | NH01 | 113.922 | 22.511 | Garden vegetation | 38.5 | 9.4 | 11.8 |
| Huaqiaocheng | HQ01 | 113.982 | 22.54 | Garden vegetation | 43.5 | 16.8 | 6.4 |
| Henggang | HG01 | 114.176 | 22.643 | Garden vegetation | 44.1 | 31.4 | 13.9 |
| Longgang | LG01 | 114.217 | 22.722 | Artificial evergreen broad-leaved forest | 38.4 | 9.2 | 11.5 |
| Meisha | MS01 | 114.296 | 22.597 | South subtropical evergreen broad-leaved forest | 38.2 | 4.4 | 22.2 |
| Yantian | YT01 | 114.236 | 22.566 | Artificial evergreen broad-leaved forest | 39.2 | 9.3 | 12.5 |
| Kuiyong | KY01 | 114.41 | 22.634 | Artificial evergreen broad-leaved forest | 33.2 | 6.8 | 22.9 |
| Nan'ao | NA01 | 114.491 | 22.538 | Artificial evergreen broad-leaved forest | 38.5 | 9.4 | 11.8 |
| Xixiang | XX01 | 113.891 | 22.58 | Garden vegetation | 35.5 | 13.5 | 11.4 |
| Lianhua | LH01 | 114.053 | 22.557 | Artificial evergreen broad-leaved forest | 44.3 | 29.1 | 7.4 |
| Tongxinling | TX01 | 114.096 | 22.545 | Garden vegetation | 26.2 | 4.1 | 5.2 |
| Honghu * | HH01 | 114.115 | 22.568 | Garden vegetation | 36.4 | 1.3 | 12.2 |
| Minzhi | MZ01 | 114.017 | 22.615 | Orchard | 38 | 12.9 | 10.2 |
| Guanlan | GL01 | 114.056 | 22.735 | Garden vegetation | 38.1 | 8.3 | 10.8 |
| Pingshan | PS01 | 114.343 | 22.711 | Garden vegetation | 43.4 | 3.1 | 18.6 |

* Since Honghu cannot be interpolated, the minimum distance method was used.

*2.3. Gaofen-4 Dataset*

In this study, the Gaofen-4 satellite, which is the first high spatial resolution remote sensing satellite in geosynchronous orbit, served as the space-based remote sensing data source. The observation range of Gaofen-4 satellites covers the whole territory of China and its surrounding areas. Furthermore, its onboard camera includes 6 wavebands, and its temporal resolution reaches up to 20 s. The visible-light/near-infrared imager has a ground resolution of 50 m, while that of the infrared payload is 400 m. The satellite scans the entire region of Shenzhen 6 to 20 times a day via pointing control. The Gaofen-4 satellite has a

spectral response function similar to the other Gaofen satellites; thus, the observed surface reflectance exhibits the same high-quality continuity and consistency as the other satellites in this series [27]. The detailed parameters of the Gaofen-4 satellite are shown in Table S1.

The Shenzhen Level-1 Gaofen-4 products were downloaded from the China Center for Resources Satellite Data Application [28]. As the accuracy of surface reflectance is a key to ensuring data accuracy, 10 scenes characterized by clear skies, reduced cloud cover, and stable observations in the region of interest were manually selected. All the 10 scenes depict an observation-time difference < 1800 s in a day (acquired at around 13:30), with no significant change in the sensor angle. Thus, the difference between sun azimuth and sensor azimuth is small.

Radiometric calibration was performed on all the images, after which fast line-of-sight atmospheric analysis of hypercubes (FLAASH) atmospheric correction was achieved using atmospheric visibility data from the ground meteorological stations. Next, rational polynomial coefficients (RPCs) information was used to carry out image rectification based on Landsat8 images captured simultaneously. The preprocessing was carried out using ENVI 5.6, and software preprocessed data was regarded as the real surface reflectance. Subsequently, cloud masks were constructed using a set of verified cloud recognition processes based on the dynamic-spectral-threshold algorithm for Gaofen-4 data [29]. Finally, the VIs were calculated using MATLAB R2020 software.

Considering the sensor's optical limitations, in this experiment, we evaluated five of the most widely used vegetation indices (Table 2). NDVI is the most widely used vegetation index to detect vegetation growth status and vegetation coverage [30]. The soil-adjusted vegetation index (SAVI) can minimize the disturbance of soil brightness, which can effectively adjust the sensitivity of soil background with incompletely covered vegetation [31]. The ratio vegetation index (RVI), also known as the simple ratio index (SR), shows a higher performance in high vegetation cover areas, compared with NDVI [32]. the two-band enhanced vegetation index (EVI2) can increase sensitivity in areas of high vegetation cover and reduce soil background and atmospheric disturbance [33]. Note that EVI2 does not use the blue band that introduces errors, and thus, it is considered superior to the three-band enhanced vegetation index (EVI1). The modified soil-adjusted vegetation index (MSAVI) is a modified version of the SAVI. MSAVI uses a self-adjustment L value and shows better performance for the vegetation sensitivity and soil noise reduction in areas with dense vegetation and complex vegetation types [34].

**Table 2.** Calculation methods of different vegetation indices.

| Index | Acronym | Formula * | Reference |
|:---:|:---:|:---:|:---:|
| normalized difference vegetation index | NDVI | $\frac{(\rho_{NIR}-\rho_R)}{(\rho_{NIR}+\rho_R)}$ | [30] |
| soil-adjusted vegetation index | SAVI | $\frac{(\rho_{NIR}-\rho_R)(1+L)}{(\rho_{NIR}+\rho_R+L)}$ | [31] |
| ratio vegetation index | RVI | $\frac{\rho_{NIR}}{\rho_R}$ | [32] |
| two-band enhanced vegetation index | EVI2 | $\frac{2.5(\rho_{NIR}-\rho_R)}{\left(\rho_{NIR}+\left(6-\frac{7.5}{c}\right)\rho_R+1\right)}$ | [33] |
| modified soil-adjusted vegetation index | MSAVI | $\frac{2\rho_{NIR}+1-\sqrt{(2\rho_{NIR}+1)^2-8(\rho_{NIR}-\rho_R)}}{2}$ | [34] |

* $\rho_{NIR}$ and $\rho_R$ are denoted as reflectance of near-infrared and red wavelengths. L = 0.5 and c = 2.

Results obtained from correlation analysis showed that the correlation coefficient was closest to the global optimal when using the NDVI, as compared to the other selected VIs. Moreover, not only was the NDVI well understood and widely applied [35], but the Gaofen-4 NDVI has also passed the consistency test with other Gaofen satellites [27]. Therefore, only the NDVI is discussed hereafter.

### 2.4. Data Availability

Clouds and cloud shadows cause errors in measuring the surface reflectance variation between two images. If the corresponding pixels on two consecutive images were not simultaneously disturbed by clouds and cloud shadows, nor covered by water, they were termed as "available pixels". When the number of "available pixels" accounted for ≥70% of all the pixels at a single station, the surface reflectance of these pixels was averaged and used for fitting. In all other cases, station data for the period were excluded and treated as missing.

It is generally believed that when rainfall intensity and precipitation reach certain levels, foliar dust is completely resuspended and removed, after which retention will begin anew [36]. If the rainfall or rainfall intensity is insufficient, foliar dust will be retained in different degrees due to differences in vegetation species [37]. From 7–15 February, Shenzhen experienced >30 mm of heavy rainfall, resulting in the complete removal of foliar dust. From 22 February–15 March, parts of Shenzhen experienced light to moderate rains. In this case, the distribution of rainfall and rainfall intensity between stations was not uniform or predictable.

Ten satellite images were collected, and nine satellite differencing images were generated. Each difference image represented changes in a period between two adjacent remote sensing observations and corresponded to atmospheric particle concentration data collected at 15 stations. As a result, 135 sets of "$\delta$NDVI - atmospheric PM10 concentration" data pairs were obtained—114 of which passed the cloud monitoring test and were found to be available pixels. However, the Nan'ao station (NA01) is too close to the sea, and the number of available pixels was often too small. In this study, NA01 had too little data available (2 out of 9) to fit, so it was excluded. In total, 112 sets of data from 14 stations were analyzed.

### 2.5. Optimal Spatial Scale between Remote-Sensing Data and Ground-Based Data

The spatial scale of the foliar dust retention process is determined by aerodynamic characteristics. That is, particles with larger sizes are more likely to be captured by vegetation surfaces [38] and to gather near the ground; they are less likely to migrate horizontally [39]. Considering that atmospheric particulate matter concentration and the dust retention effect have spatial autocorrelation, we used the average value of $19 \times 19$ pixels (i.e., $950 \times 950$ m) around the station to improve the fitting accuracy, since all 5 VIs achieve maximum correlations around this point (Figure 3). Zhao et al. [10] reported that dust from coal mines in Inner Mongolia has a radius of influence of approximately 900 m on the surrounding grassland canopy, while the range of maximum dust retention is 300–500 m. Considering that trees can also shorten the particles' migration distance by reducing near-surface wind speed [7], we found that the maximum correlation radius in this study was 425 m, which is consistent with the results of aforementioned study.

### 2.6. Theoretical Basis for Correlation between PM10 Concentration and $\delta$NDVI within a Station

Previous spectral experiments have shown that the five VIs used in this study were all linearly or logarithmically correlated with foliar dust retention [35,40], which is the theoretical basis for the potential correlations between NDVI and EEFDR.

The relationship between atmospheric PM10 concentration and foliar dust retention is mathematically expressed as follows [22]:

$$F_p = V_d \times C_p \tag{1}$$

where $F_p$ is the mass particle flow rate ($\mu$g s$^{-1}$m$^{-2}$) towards the leaf surface; $V_d$ represents the deposition velocity (m s$^{-1}$) as a measure of filtration performance, which is a dynamic parameter determined by factors, such as vegetation physiological characteristics, meteorological conditions, and particle physicochemical characteristics; and $C_p$ is the atmospheric particle concentration ($\mu$g s$^{-1}$m$^{-3}$).

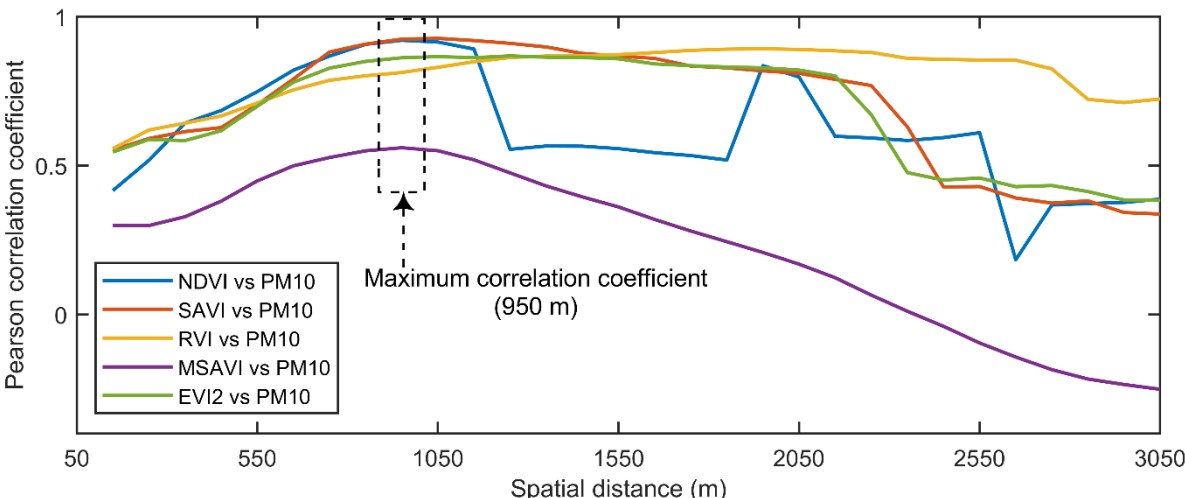

**Figure 3.** Variations in Pearson correlation coefficients of vegetation indexes versus atmospheric PM10 concentration with different sampling distances. Note that the maximum correlation coefficient occurs at the spatial distance of 950 m.

In this study, we expected that $F_p$ was linearly correlated with the δNDVI; $C_p$ can be represented by the daily atmospheric PM10 concentration obtained from the air pollution monitoring stations, and $V_d$ was closely related to the vegetation characteristics. As the vegetation composition and structure were considered unchanged at a given station within the study period, $V_d$ was treated as a constant. According to Equation (1), the δNDVI impacted by the dust retention effect would be linearly correlated with the atmospheric PM10 concentration at the same station during this period.

## 3. Results

### 3.1. Space-Dependent δNDVI-PM10 Relationship

We analyzed the correlation between PM10 and δNDVI based on all samples. NDVI is expected to be responsible for the differences in fitting between different stations. Linear regression analysis was performed on the data of δNDVI and atmospheric PM10 concentration for data from all stations and from each station (Figure 4). Table S2 shows the original and fitting curve data for each station, including their root mean squared error (RMSE), correlation coefficient (PCC$_{(δNDVI-PM10)}$), and regression slope (RS$_{(δNDVI-PM10)}$). All the samples in each station have passed the Shapiro–Wilk parametric hypothesis test. Therefore, PCC is used to measure data correlation within the same station. Given the skewed distribution of the cross-station PM10 dataset, we used the Spearman correlation analysis on the overall dataset to verify the cross-station correlation of PM10-δNDVI. There are significant correlations between the δNDVI and the atmospheric PM10 concentration for data from all stations. For individual stations, 8 out of 14 stations showed significant correlations.

The NDVI significantly correlated with the PCC$_{(δNDVI-PM10)}$ ($n = 14$, $p < 0.001$; Figure 5a) and correlated with the RS$_{(δNDVI-PM10)}$ across stations ($n = 14$, $p < 0.05$; Figure 5b). This result demonstrates that the site-specific PCC$_{(δNDVI-PM10)}$ and RS$_{(δNDVI-PM10)}$ were both significantly related to the site-specific NDVI. That is, a station with a smaller NDVI is likely to have a more significant correlation and a more negative regression slope of δNDVI versus atmospheric PM10 concentration. The RS$_{(δNDVI-PM10)}$ for Yantian station (YT01) was much smaller ($-1.73 \times 10^{-5}$) than those of all the other stations (range of $-9.53 \times 10^{-4}$ to $-1.43 \times 10^{-4}$).

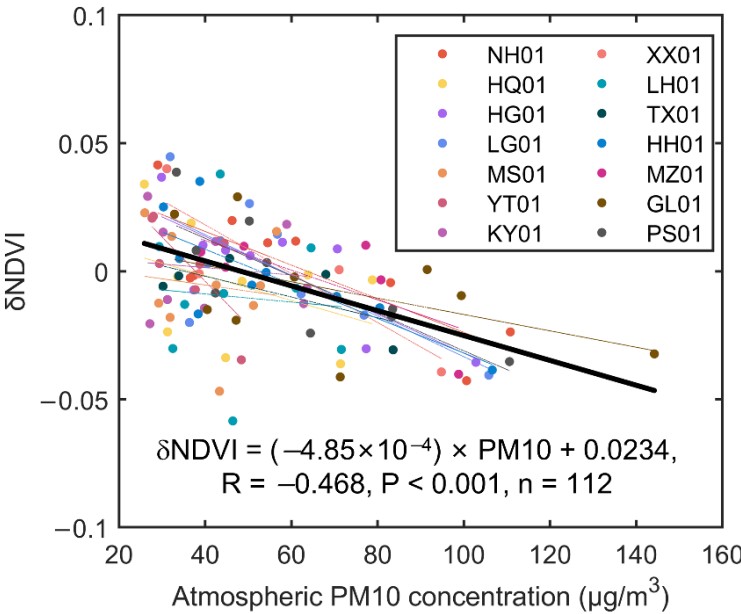

**Figure 4.** Correlation between atmospheric PM10 concentration and the relative change in the normalized difference vegetation index (δNDVI) at all stations over the study period. Each color represents a dataset for a station. Each point represents one data pair in one period, and each line represents the fitting curve of one station during the entire period (black line, $R = -0.468$, $n = 112$, $p < 0.001$). The thick black line represents the fitting line for all 112 data. See Figure S1 for individual fitting of each station.

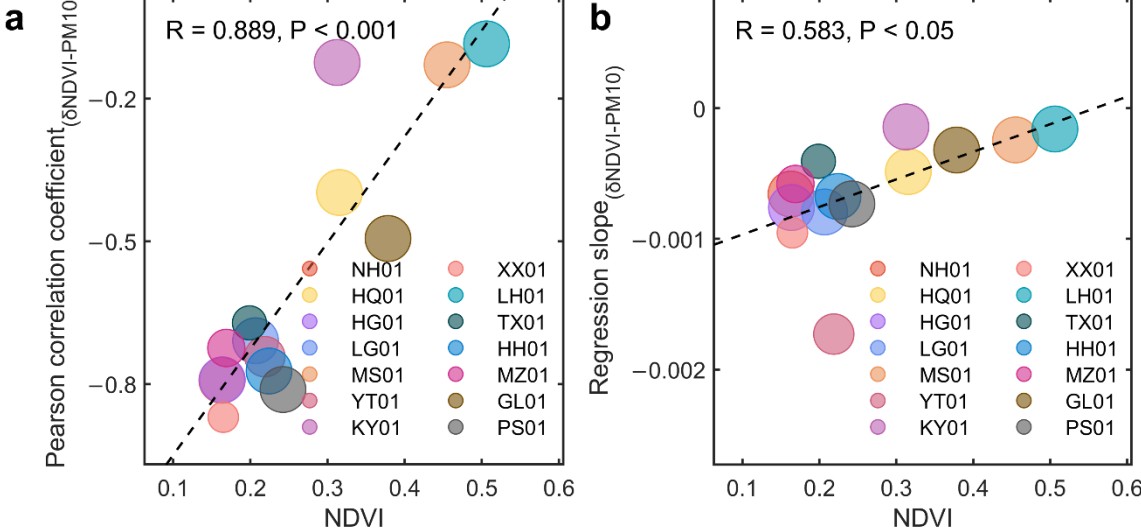

**Figure 5.** Normalized difference vegetation index (NDVI)-dependent δNDVI-PM10 relationship. (**a**) Fitting between NDVI and Pearson correlation coefficient of δNDVI versus atmospheric PM10 concentration. (**b**) Fitting between NDVI and regression slope of δNDVI versus atmospheric PM10 concentration. The radius of the solid circle represents the number of available data values at a station. The radius of the solid circle represents the number of available data at a station.

*3.2. Time-Dependent δNDVI-PM10 Relationship*

We detected the significant change of correlation in continuous time influenced by rainfall events. Figure 6 shows the data pairs and their frequency histogram of δNDVI and atmospheric PM10 concentration in the three different meteorological scenarios: before, during, and after rainfall. Compared with data from periods 1–5 (before rainfall), data from

periods 6–9 (during and after rainfall) showed obvious deviations, i.e., a greater increase in δNDVI during rainfall and a larger decrease in δNDVI after rainfall. Two fitting lines of periods 1–9 and periods 1–5 (excluding data from periods 6–9, which were biased by rainfall) are also shown. Compared to the Spearman's R based on all data (periods 1–9; $R = -0.468$, $n = 112$, $p < 0.001$), Spearman's R based on data from periods 1–5 was significantly improved ($R = -0.680$, $n = 63$, $p < 0.001$).

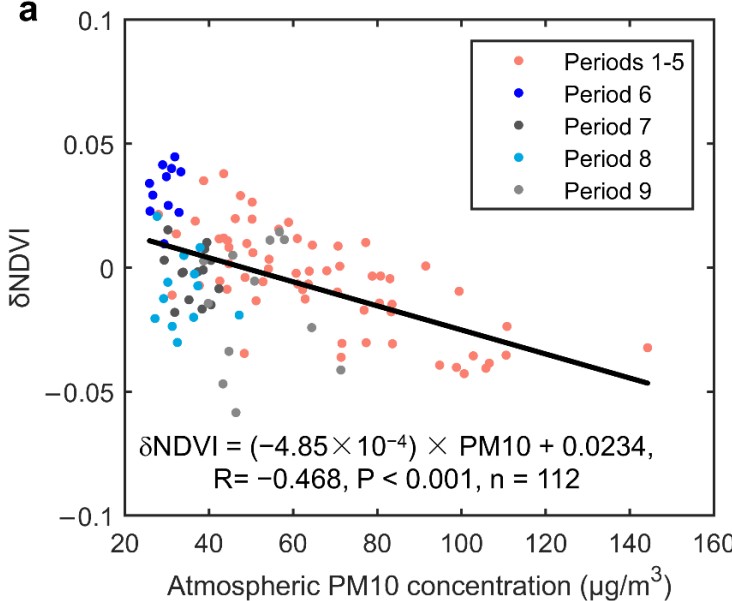

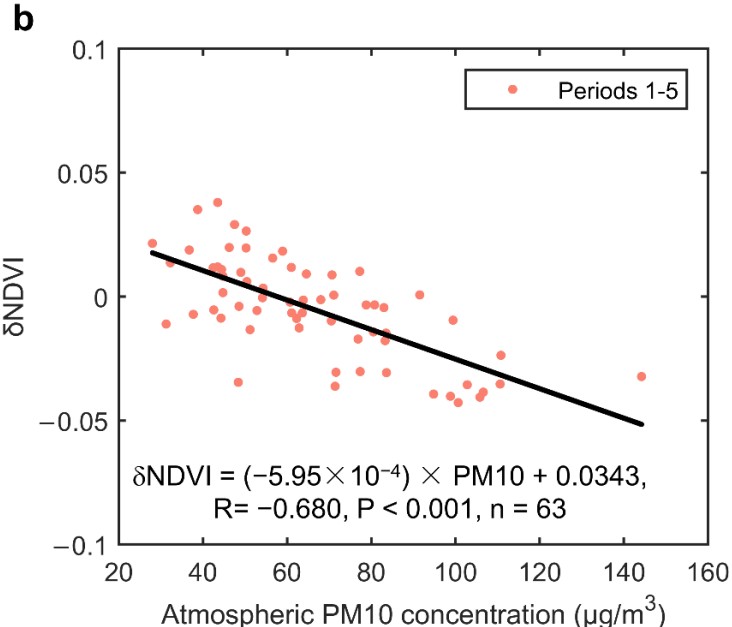

**Figure 6.** Scatter diagram of atmospheric PM10 concentration and the variation rate of the normalized difference vegetation index (δNDVI) at 14 stations over the study period, along with their fitting curves. (**a**) Points and fitted line of data from period 1 to 9 (black line, $R = -0.468$, $n = 112$, $p < 0.001$). This highlights the data pairs from period 6 (dark blue), period 7 (dark gray), period 8 (light blue), and period 9 (light gray), which deviate from the other data due to the influence of rainfall. A period refers to a time interval between two adjacent remote sensing observations. Note that data from period 8 and period 9 were influenced by uneven rainfall (See Figure 2 for specific data). (**b**) Points and fitted line of data from period 1 to 5 (salmon pink line, $R = -0.680$, $n = 63$, $p < 0.001$).

## 4. Discussion

### 4.1. NDVI-Dependent Performance in Estimating δNDVI by PM10

As PCC is the commonly used statistic for the goodness of fit, the significant negative correlation between $PCC_{(\delta NDVI-PM10)}$ and NDVI suggests that the δNDVI could be more accurately estimated by PM10 in areas with low NDVI. This could be attributed to three possible reasons: First, densely planted trees reduce air exchange near the surface. Thus, particles are mostly deposited on the forest edges and only partially enter the forest interior at high wind speeds [41]; foliar dust retention in the inner forest canopy is affected to a greater extent by factors such as wind speed, as opposed to atmospheric particle concentration, which may partially contribute to the less significant δNDVI-PM10 correlations in forests with high NDVI. Second, during the growing season, the NDVI in densely vegetated areas is mainly regulated by plant phenological factors instead of EEFDR. However, the areas with sparse vegetation are mostly urban, and their NDVI may be more influenced by the anthropogenic PM10 emission source, which may contribute to the more significant δNDVI-PM10 in urban areas with low NDVI. Third, foliar dust can settle on the overall plant surface. As such, the total leaf surface should be considered instead of just the leaf surface observed by remote sensing. Beckett et al. [7] suggested that using estimated LAI instead of real LAI values may result in a 10-fold difference in total foliar dust estimates. This is manifested here as a decrease in correlation between EEFDR and observable leaf surface. Thus, the higher the NDVI, the more complex the vegetation community structure and stratification, and the bigger the total leaf surface, resulting in less significant δNDVI-PM10 correlations.

$RS_{(\delta NDVI-PM10)}$ is also significantly positively correlated with the NDVI, possibly due to two reasons. First, the forest edge blocks the free diffusion of particles through the forest canopy and slows EEFDR in the forest interior, resulting in smaller absolute $RS_{(\delta NDVI-PM10)}$ values, i.e., a lower sensitivity of δNDVI to PM10, in a forest with high NDVI. Second, atmospheric humidity changes the size of atmospheric particles. That is, when humidity rises, the particles absorb water and become larger, and are more easily captured by the leaf surface [38]. Hanel [42] compared the deposition velocities of three aerosol types under different humidity levels and found a clear exponential relationship between the deposition velocity and relative humidity. Areas with dense vegetation usually have higher humidity than areas with sparse vegetation. This local meteorological difference may be one of the reasons why $RS_{(\delta NDVI-PM10)}$ increases with the NDVI. We also noted that station YT01 yielded a distinctly lower $RS_{(\delta NDVI-PM10)}$ value and lower atmospheric PM10 concentrations compared to all the other stations. Chen et al. [43] proposed that on coasts with high humidity and low wind speeds, a government-controlled air-monitoring station will significantly underestimate the true value of PM10. Therefore, the distinct $RS_{(\delta NDVI-PM10)}$ value in YT01 may be attributed to the fact that YT01 is a station that is adjacent to the coast (<800 m from the shoreline). Excluding the YT01 station from the whole dataset, the PCC between $RS_{(\delta NDVI-PM10)}$ and NDVI rose from 0.583 to 0.799, implying that the EEFDR of stations or areas similar to YT01 may not be well predicted by a general δNDVI–PM10 relationship.

### 4.2. Three Scenarios of Spectral Characteristics of Foliar Dust Retention

Foliar dust retention increases gradually over time, but does not depict a linear relationship with time, because it is frequently disturbed by meteorological factors [44]. During the dust retention process, the following three scenarios are alternately observed:

Scenario A: before rainfall—fluctuation of the maximum dust retention during the dry period: The Spearman's R reaches 0.680 ($p < 0.001$) for data from periods 1–5, indicating a significant correlation between δNDVI and PM10. Note that there are multiple points (25 out of 63) where the δNDVI is greater than 0; this is because once the maximum dust retention is reached (usually within 15 days of rainfall), foliar dust retention will no longer increase, but fluctuate with the ambient PM10 atmospheric concentration. The maximum dust retention is defined as the state where the sum deposition velocity reaches zero. Due

to the simultaneous deposition and resuspension of foliar dust, the sum deposition velocity is dynamic [45], and can be expressed as:

$$|V_{sum}| = |V_{dep}| - |V_{res}| \tag{2}$$

where $V_{sum}$, $V_{dep}$, and $V_{res}$ represent the sum deposition velocity, net deposition velocity to the leaf surface, and resuspension velocity flowing out of the leaf surface, respectively. According to Equation (2) $V_{res}$ is positively related to foliar dust accumulation [46] and is one of the main reasons for the variability associated with sum deposition velocities [22].

The first rainfall in this experiment occurred 116 d after the previous rainfall, on October 17 of the previous year. Therefore, periods 1–5 were included in scenario A. In this scenario, when PM10 is high, the $V_{sum}$ is greater than 0, and the maximum dust retention increases, resulting in a more significant EEFDR and negative δNDVI, and vice versa.

Scenario B: during rainfall—foliar dust retention being washed off by rainfall: A rainfall event occurs accompanied by prolonged periods of cloudy days, which are not conducive for remote sensing observations. For instance, in this study, 2 d before and 4 d after the rainstorm on February 10 were all cloudy days in Shenzhen. Therefore, the during-rainfall period inevitably included several cloudy days before and after rainfall. Nevertheless, the NDVI rose sharply in period 6, indicating that the cleaning effect of rainfall on the foliar dust still dominated period 6. Note that period 8 contained two localized rainfalls that were not strong, which resulted in a generally weak and uneven cleaning effect. The cloudy weather also lasted longer (i.e., 3 d before the first rainfall, and 8 d after the second rainfall), which resulted in a long-term dust retention process after the rain, partially offsetting the rising of NDVI, hence producing a lower δNDVI value than period 6.

Scenario C: after rainfall—net accumulation of foliar dust retention on clean leaves In this scenario, when exposed under the same atmospheric PM10 concentration, the NDVI would be reduced faster than in scenario A, i.e., the absolute value of the negative δNDVI in scenario C would increase more than that in scenario A. Foliar dust is washed away during rainfall events and will resettle after the rain stops. According to Equation (1), the deposition velocity of dust should be proportional to the atmospheric particle concentration. Foliar surfaces will rapidly accumulate dust until they reach their maximum capacity. Wang et al. [47] reported that it takes, on average, 3 weeks for 19 species of common plants in Suzhou City to reach maximum dust accumulation capacity. This scenario is reflected in the data deviation observed after rainfall events—i.e., periods 7 and 9—as compared with data collected during other times throughout the experiment. Note that data points from period 9 are scattered because the rainfall intensity and precipitation in period 8 did not meet the requirements of complete scouring; thus, the degree of scouring of foliar dust at each station was different.

Therefore, excluding the during- and after-rainfall data (i.e., periods 6–9 here) improves the goodness of fit between δNDVI and PM10 (from $R = -0.468$, $n = 112$, $p < 0.001$ to $R = -0.680$, $n = 63$, $p < 0.001$) (Figure 6).

### 4.3. Potential for EEFDR Modeling

The high goodness of fit between δNDVI and PM10 ($R = -0.680$, n = 63, $p < 0.001$) not only confirms the observability of EEFDR by remote sensing, but also provides a theoretical basis for the continuous simulation of EEFDR using PM10 data in natural meteorological scenarios. To the best of our knowledge, this is the first study using a high spatitemporal resolution satellite to examine the correlation between δNDVI data and atmospheric PM10 data based on the extinction effect of foliar dust retention on NDVI. In terms of the theoretical relationship between foliar dust retention and vegetation spectrum, the high goodness of fit is consistent with our expectations.

In our study, for the first time, PM10 as a proxy of dust was used instead of foliar dust retention sampling. The manual sampling and weighing of foliar dust have numerous disadvantages, including high cost, limited number of sampling points, and difficulty in matching the data with remote sensing data at different spatial scales. Moreover, diverse

sampling and experimental processes and conditions make it difficult to compare results among different studies. Previous studies on foliar dust retention spectral characteristics were mostly carried out in the laboratory at the leaf level instead of with aerospace remote sensing at the canopy level: Between these two levels, the dust retention effect transformation involves canopy attributes, dorsiventral leaf reflectance properties, and the vegetation community's morphology, among factors [48–50]. Furthermore, using PM10 data also avoids the difference in spectral response between different sensors [10] and the interpretation of mixed pixels [51].

Previous studies suggested that the difference between two remote sensing images could be regarded as EEFDR under similar weather backgrounds. Ma et al. [35] used 2 Landsat images taken 11 years apart, Yan [13] used 3 MODIS images obtained several months apart, and Kayet [50] used 2 images each from Landsat and Hyperion taken 11 years apart. Similar approaches have been applied to other satellites (e.g., Sentinel-2, EO-1) to simulate foliar dust retention [51–54].

In contrast, we used 10 images within 90 days. Such high temporal resolution of observations not only reduces possible interference to NDVI from phenology and LUCC, but also makes it possible to observe continuous changes in EEFDR. This is because foliar dust retention fluctuates wildly with meteorological events, such as precipitation or high winds, which will cause significant shedding of foliar dust [40,44]. Observations too far apart will not be able to distinguish these meteorological events.

More importantly, all of the above experiments assume that the EEFDR in the remote sensing image are equal to the difference between the two remote sensing observations. This assumption is doubtful and shaky, especially considering that two observations were obtained more than years apart. Our method can be applied to circumstances where this assumption does not apply.

Variable influence of vegetation species and vegetation topography will also lead to differences in dust retention capacity [55]. This may be one of the important reasons for the different PM10-δNDVI correlations among different stations. However, the good correlation between different stations proves that our method can resist such disturbance. All tree species in the experimental area are evergreen broad-leaved species, which minimizes seasonally-driven leaf shedding and growth, which is an advantage of studying foliar dust retention in low-latitude regions. Species differences should be one of the future research priorities in high-latitude regions.

Although the temporal resolution of the Gaofen-4 satellite reaches 20 s, the average interval for obtaining cloud-free images is still approximately 9 d, and it is limited by the weather conditions in the study area. Observations would be even more challenging in the rainy season, which makes it almost impossible to quantify the EEFDR on and after the day of rainfall.

The Gaofen-4 satellite used in this study has only six bands, which may cause some important spectral features to be omitted [38]. However, our research targets NDVI rather than hyperspectral data, which is sufficient for general satellite remote sensing. Zhao [10] used hyperspectral drones to retrieve the distribution of dust retention in a grassland based on random forest regression. Continuous joint observations of satellites and drones may be a future feasible method for a more complete and accurate quantification of dust retention and vegetation indices.

## 5. Conclusions

Based on the Gaofen-4 satellite imagery dataset, the atmospheric PM10 concentration dataset, and the meteorological station dataset, we checked the correlation between NDVI and PM10 to estimate the extinction effect of foliar dust retention on urban vegetation. We quantified the correlations between δNDVI and atmospheric PM10 concentration, which were expected to not only show NDVI-dependency, but also vary under different meteorological scenarios. The main conclusions drawn as a result of this study are as follows:

(1) The use of PM10 data can replace artificial sampling and laboratory experiments, which creates the possibility for the large-scale inversion of EEFDR.

(2) Dense vegetation will weaken the correlation between PM10 and EEFDR.

(3) The effect of rainfall on foliar dust retention must be taken into account; thus, a complete foliar dust retention process should include three different meteorological scenarios, which is the premise for the continuous observation of EEFDR.

This study demonstrates the direct influence of EEFDR on remote sensing NDVI observation and provides new possibilities for estimating the regional EEFDR, building multi-factor EEFDR models, and correcting the vegetation indices based on remote sensing in the future.

**Supplementary Materials:** The following supporting information can be downloaded at: https://www.mdpi.com/article/10.3390/rs14205103/s1, Table S1. Main technical indexes of the Gaofen-4 camera. Table S2. Detailed information of all stations' data and fitting lines. Latitude and longitude are detected in the WGS84 coordinate system; plants are all evergreen. Figure S1. Fitting curves of 14 stations between their atmospheric PM10 concentrations and the normalized difference vegetation index (δNDVI). They were separated from Figure 4.

**Author Contributions:** Conceptualization, T.Y. and J.W.; methodology, T.Y.; software, T.Y.; validation, T.Y.; formal analysis, T.Y.; resources, T.Y. and Y.C.; data curation, Y.C.; writing—original draft preparation, T.Y.; writing—review and editing, J.W.; visualization, T.Y.; supervision, H.Z.; funding acquisition, H.Z. All authors have read and agreed to the published version of the manuscript.

**Funding:** This work was supported by the Shenzhen Fundamental Research Program (No. GXWD202 01231165807007-20200812142216 001), to which we are very grateful.

**Data Availability Statement:** The remote sensing dataset was obtained in 17 May 2021 from the China Center for Resources Satellite Data and Application and are available from http://www.cresda.com with the permission of the China Center for Resources Satellite Data and Application; the meteorological dataset was obtained in 22 May 2021 from the Meteorological Bureau of Shenzhen Municipality and are available from http://weather.sz.gov.cn with the permission of the Meteorological Bureau of Shenzhen Municipality.

**Conflicts of Interest:** The authors declare no conflict of interest.

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
