# Peer review of "Extinction Effect of Foliar Dust Retention on Urban Vegetation as Estimated by Atmospheric PM10 Concentration in Shenzhen, China"

_remotesensing, doi:10.3390/rs14205103_

Round 1

Author Response

We sincerely thank the reviewer for these comments. We have carefully considered each of the reviewer’s concern and made tremendous revisions. Please see the attachment.

Reviewer 2 Report

The article sent for review dealt with the topic of extinction effect of foliar dust retention on urban vegetation as estimated by atmospheric PM10, an example of a city in China. The introduction to the literature was presented correctly and the reasoning for posing the research problem sufficient. The experiment was conducted correctly and the indication of correlation between the studied properties of satellite images - NDVI coefficient, δNDVI, and PM10 dust demonstrated.

However, according to the conclusions, this research does not bring groundbreaking knowledge for science no less it increases the scope of information on the subject and indicates further development on this topic.

In terms of specific elements for improvement:

Fig 1 China is not a planet, there are other countries that border China, please improve the map.

Fig 3 the legend obscures part of the chart

I rate the article as well as the research well prepared and executed however average in its form.

Author Response

(The authors gave the same response as above.)

Reviewer 3 Report

Dear Authors,

I like very much your idea and the research topic, but the current version of the manuscript should be significantly improved, because your manuscript looks like a report of an experiment.

You have a lot of well-known statements, so you need to add much more details presenting much better theoretical background, how different authors solve the research problem. You need to add much more details to all parts of the manuscript, but the most important is an accuracy assessment and validation of used models.

it is not clear what is your input data? how did you test and verified them?

Discussion needs much more direct comparisons between your more important results and achievements of different authors.

Much more detailed comments you can find in the attached manuscript.

Best regards

Reviewer

Author Response

(The authors gave the same response as above.)

Reviewer 4 Report

The article is interesting and could make an important contribution to the field, but unfortunately in its current form the manuscript lacks research depth, visible by a focus on the case study rather than the research issue, proved by poor introduction and discussions. Thus, the manuscript requires a strong development of these sections. Moreover, the manuscript is poorly organized, and requires moving elements to the sections where they belong. Detailed comments are provided for each section of manuscript.
In a scientific article the introduction has the role of reviewing existing literature in order to identify its shortcomings (misconceptions, lacks, errors, uncertainties etc.) and declaring the research goals, emphasizing their contribution to addressing the previous shortcomings, as well as the novelty of the study. In this article, the introduction lacks the most important part, which is the declaration of the research goals. Perhaps they exist, but the way they are written, without any clear statement (e.g., "this study aims to..."), gives the impression that they are missing. The research goals must be phrased clearly, using a strong statement. Also, the introduction ends with an absolutely unnecessary and confusing outline of the structure of article or research steps. This paragraph should be removed, as an article is summarized by its abstract and does not need any other summary. Instead it would help pointing out, after declaring the research goals, the novel and original elements of the current research. The introduction should be cleaned of elements not belonging here, such as the methods used in the study, results, or their significance.
Figure 1 shows the inability of authors to write up research. This is an article for an international journal, and not a report for the national authorities. The authors should present a map showing the location of the study area in an international context, making visible the neighboring countries with their names, so that a Brazilian researcher could understand it too. China is not the only country in the world!
The most important section of a research article, the Discussions, is missing. The section is meant to emphasize the importance of research, justifying its publication. Normally, this section includes include (A) the significance of results - what do they say, in scientific terms; (B) the inner validation of results, against the study goals or hypotheses; (C) the external validation of results, against those of similar studies from other countries, identified in the literature; (D) the importance of results, meaning their contribution (conceptual or methodological) to the theoretical advancement of the field; (E) a summary of the study limitations and directions for overcoming them in the future research. Only the significance of results is presented. The "Discussions" should be developed to include the missing elements.
Conclusions are not sufficiently broad in scope, and lack research depth, pertaining only to the case study and being in fact just a summary of the main findings. Conclusions are meant to deliver a scientific message, far away beyond the case study, to the entire scientific community, making a clear contribution to the theoretical (conceptual or methodological) development of the field. Conclusions must be developed beyond the case study.

Author Response

(The authors gave the same response as above.)

Round 2

Reviewer 1 Report

1.How to prove that seasonal growth in the region from January to April does not affect the value of NDVI?

2.Line 215,different vegetation has different dust retention capacity. The dust retention capacity of vegetation itself can affect the correlation between NDVI and pm2.5. How to exclude this factor interference?

3.At a certain monitoring site, if the NDVI value is relatively high and PM10 is also high, is it the effect of the dust retention capacity of the plant itself. How to consider this point?

Author Response

We sincerely thank the reviewer for this comment. We completely agree with the reviewer's concerns that NDVI always changes with phenology, regardless of plant species. The more phenology affects NDVI, the more blurred the correlation between NDVI and PM10 will be.

We used January 1 to April 1 as the study period. This is not because we believe that phenology does not affect NDVI during this period, but because we want to reduce the effect of phenology on NDVI to highlight the extinction effect of foliar dust retention (EEFDR) on NDVI.

The research goal is not to demonstrate whether there is a correlation between PM10 and EEFDR in theory, but to test whether such a relationship can be observed by satellite remote sensing in nature.

While we agree that there must be phenological interference in the PM10-NDVI relationship, there is still a significant correlation between PM10 and δNDVI. This justifies that PM10 and dust retention are much more important factors than phenology in nature.

Reviewer 3 Report

Dear Authors,

the manuscript still looks like a report of a conducted experiment, because you described all your steps, but there is a lack of the theoretical background and the Discussion looks like an analysis of scored results, you need to compare directly your more important result by result with references.

Most of the Introduction, description of used methods and Discussion is still too general, you need to add much more details.

So, concluding, I can see a progress in revising of your manuscript, but it is still a halfway to a version, which could be accepted.

Best regards

Reviewer

Reviewer 4 Report

Despite my clear comments, indicating that the role of introduction is to critically analyze previous studies in order to declare the research goals, and NOT to summarize the study (which is done by the abstract), the revised version of the article reads in the final paragraph of introduction (line 84) "The research contents of this study include...", which a summary of research, NOT belonging to the introduction. If in fact the research questions or goals are following, the sentence should read "the study goals / research questions are..."

Author Response

We sincerely thank the reviewer for this comment. We completely agree with the reviewer that the research goals, instead of a research summary, should be clearly stated in the Introduction. We apologize for our inappropriate wording. Based on the reviewer’s suggestion, we have modified the last paragraph of the Introduction in the latest version of the manuscript:

The study aims to: (1) quantify information on the potential correlation between δNDVI - an indicator of EEFDR, and atmospheric PM10 concentration - an indicator of foliar dust retention; (2) identify the space-dependency of the correlation based on site-specific NDVI; and (3) identify the time-dependency of the correlation based on different meteorological scenarios. (Lines 89-93)